# Efficacy of Combined Oral Isotretinoin and Desloratadine or Levocetirizine vs. Isotretinoin Monotherapy in Treating Acne Vulgaris: A Systematic Review and Meta-Analysis of Randomized Controlled Trials

**DOI:** 10.3390/biomedicines13081847

**Published:** 2025-07-30

**Authors:** Julia Woźna, Andrzej Bałoniak, Jan Stępka, Adriana Polańska, Ewa Mojs, Ryszard Żaba

**Affiliations:** 1Department of Dermatology and Venereology, Poznan University of Medical Sciences, 60-355 Poznań, Poland; 2Doctoral School, Poznan University of Medical Sciences, 61-701 Poznań, Poland; 3Clinical Hospital of the Poznań University of Medical Sciences, Poznan University of Medical Sciences, 61-701 Poznań, Poland; 4Department of Clinical Psychology, Poznan University of Medical Sciences, 61-701 Poznań, Poland

**Keywords:** acne vulgaris, retinoids, isotretinoin, antihistamines, desloratadine, levocetirizine

## Abstract

**Background/Objectives:** Acne vulgaris is a widespread, chronic inflammatory skin condition that significantly impacts patients’ quality of life. Although oral isotretinoin remains the most effective treatment, recent evidence suggests that H_1_-antihistamines such as desloratadine and levocetirizine may enhance acne therapy. This study assesses whether combining H_1_-antihistamines to isotretinoin enhances treatment efficacy in acne vulgaris compared to isotretinoin alone. **Methods:** Our analysis included 10 randomized controlled trials involving 675 patients collectively, predominantly from Asia and the Middle East. Data were extracted by two independent reviewers, with discrepancies resolved by a third. Risk of bias was assessed using the Cochrane RoB 2 tool. Analyses were performed using RevMan 5.4 with random-effects models, and heterogeneity was evaluated via I^2^ and Q tests. Sensitivity analyses were conducted to assess result robustness. **Results:** Combination therapy with isotretinoin and desloratadine showed a significantly greater reduction in GAGS (Global Acne Grading Scale) score by week 12 (*p* < 0.00001; MD 2.68, 95% CI 1.60 to 3.75; I^2^ = 0%) while earlier timepoints showed non-significant or borderline results. For inflammatory lesions, significant improvements with desloratadine emerged at weeks 4, 8, and 12 after excluding an influential outlier, with low heterogeneity and consistent direction of effect. Non-inflammatory lesions did not differ significantly at weeks 4 or 8. At week 12, a significant reduction was seen in the desloratadine subgroup (OR 2.61, *p* = 0.003, I^2^ = 11%) and in overall pooled analysis (OR 2.77, *p* < 0.0001, I^2^ = 2%). Among side effects, acne flare-ups, pruritus, and cheilitis were significantly reduced in the desloratadine group, as well as in pooled analysis. Xerosis did not consistently differ between groups. Overall, desloratadine improved tolerability and reduced mucocutaneous adverse events more than levocetirizine. **Conclusions:** Current evidence suggests that combining oral antihistamines with isotretinoin may offer therapeutic benefits in acne management, particularly in enhancing tolerability and potentially improving clinical outcomes, as reflected by significant reductions in GAGS scores and mucocutaneous adverse effects such as cheilitis, pruritus, and acne flare-ups.

## 1. Introduction

Acne vulgaris is a common skin disease that affected 9.4% of the global population in 2010, making it the eighth most prevalent condition worldwide [1]. It is a chronic, inflammatory disorder of the pilosebaceous unit, typically presenting with open or closed comedones, papules, pustules, or nodules on the face or trunk. These lesions can lead to pain, erythema, hyperpigmentation, and scarring [2]. The severity and chronic nature of acne can significantly impair patients’ quality of life and may lead to depression, anxiety, low self-esteem, and even suicidal ideation [3]. In fact, acne ranked second among all skin diseases in terms of disability-adjusted life years (DALYs) in 2017 [4].

According to the American Academy of Dermatology (AAD), strong recommendations exist for the use of benzoyl peroxide, topical retinoids, and topical antibiotics, as well as oral doxycycline in the management of acne. Oral isotretinoin is strongly recommended for severe acne, acne causing psychosocial burden or scarring, or acne that is unresponsive to standard oral or topical therapies [5]. Isotretinoin is an effective treatment that can yield excellent therapeutic outcomes in patients with severe acne. It has been the only FDA-approved treatment for severe recalcitrant nodular acne vulgaris since 1982 [5]. However, its use is often accompanied by adverse effects, particularly affecting the mucocutaneous, musculoskeletal, and ophthalmic systems.

Given these challenges, current research is exploring adjunctive therapies to improve treatment outcomes and reduce adverse effects in patients who poorly tolerate isotretinoin [6]. 

H_1_-antihistamines like desloratadine and levocetirizine, known for their excellent safety profile, are of particular interest. Beyond their anti-inflammatory, mast cell stabilization, and anti-chemotactic effects, studies have shown that sebocytes express H_1_ receptors and blocking these receptors can reduce squalene production, which is a key component of sebum [7]. This suggests that antihistamines may help modulate sebum output, offering a dual benefit in acne treatment when combined with isotretinoin.

Previous reviews [8,9,10] evaluated this therapeutic group based on a limited number of studies and fewer outcomes. With an increasing number of recent trials, there is a clear need to update and critically appraise the latest evidence on combination therapy involving isotretinoin and antihistamines. A comprehensive assessment of clinical outcomes, study design, and statistical data is needed to better determine its therapeutic value. Considering the global impact of acne and the common use of isotretinoin, improving treatment strategies to support adherence and effectiveness remains clinically important [6]. Given these reasons we aimed to analyze the available randomized controlled trials (RCTs) and assess the efficacy of combined oral isotretinoin with desloratadine and oral isotretinoin with levocetirizine versus isotretinoin alone in the treatment of acne vulgaris.

## 2. Materials and Methods

This review was conducted in accordance with the PRISMA 2020 guidelines and the Cochrane Handbook for Systematic Reviews and Meta-Analyses and was prospectively registered in PROSPERO on 20 April 2025 (CRD420251036678).

### 2.1. Eligibility Criteria

Inclusion criteria were based on the PICO format. The study population (P) included patients diagnosed with acne vulgaris. Interventions (I) involved treatment with isotretinoin combined either with desloratadine or levocetirizine, compared to isotretinoin alone (C). Studies were required to report at least one of the predefined outcomes (O), including Global Acne Grading Scale (GAGS), cumulative incidence of side effects (e.g., xerosis, pruritus), or total inflammatory and non-inflammatory lesions count. Only randomized controlled trials (RCTs) were eligible for inclusion, with no restrictions on follow-up duration. Exclusion criteria included: non-English publications, case reports, reviews, non-comparative studies, and trials with fewer than 10 participants per treatment arm. 

### 2.2. Search Strategy

A comprehensive literature search was conducted across PubMed (MEDLINE), Cochrane Library, and Embase from inception to April 20th, 2025. Search terms included: “Desloratadine”, “Levocetirizine”, “Isotretinoin”, and “Acne vulgaris”, combined using Boolean operators “AND” and “OR”. A full research strategy is available in Appendix A. Reference lists of included studies and previous meta-analyses were hand-searched to identify additional eligible trials. 

### 2.3. Extracted Variables

The following data were extracted: first author’s name, publication year, country, study design, sample size, age and sex distribution, isotretinoin dosage, type of antihistamine used, mean change and standard deviations in GAGS score from baseline to given follow-ups, mean change and standard deviations in inflammatory lesion count from baseline to given follow-ups, and reported side effects. Standard deviations for mean change were calculated based on an assumed correlation coefficient (r = 0.5), in accordance with the Cochrane Handbook for Systematic Reviews of Interventions. In cases where numerical data were not available in the tables, relevant figures were identified and analyzed using WebPlotDigitizer (version 5) to extract the required data.

### 2.4. Data Collection Process

Two independent reviewers (J.W. and A.B.) conducted the selection process. After duplicate removal using Zotero (version 7), titles and abstracts were screened, followed by full-text screening. Discrepancies were resolved by consensus, or by consultation with a third author (J.S.).

### 2.5. Endpoints

Primary outcomes were defined as mean change in GAGS score from baseline to individual follow-ups. Secondary outcomes included a change in inflammatory and non-inflammatory lesion count at individual follow-ups and cumulative incidence of common side effects such as acne flare-ups, xerosis, cheilitis, and pruritus. 

### 2.6. Risk of Bias Assessment

Two independent reviewers (J.W. and A.B.) assessed the risk of bias in all included randomized controlled trials with the Cochrane Risk of Bias 2.0 tool (RoB 2), which covers five domains: (1) randomization process, (2) deviations from intended interventions, (3) missing outcome data, (4) outcome measurement, and (5) selection of the reported result. Any disagreements were adjudicated by a third author (J.S.). Publication bias was assessed using funnel plots to identify potential asymmetry and outlier studies. 

### 2.7. Statistical Analysis

Statistical analysis was performed in RevMan software v5.4 (Cohrane Collaboration). Continuous outcomes were pooled using mean differences (MDs) and odds rations (ORs) with 95% confidence intervals (CIs). Dichotomous outcomes were analyzed using ORs with 95% CIs. Heterogeneity was evaluated using the I^2^ statistic and the Cochran Q test, with I^2^ > 50% and *p* < 0.10 considered indicative of substantial heterogeneity. All analyses were conducted with a conservative approach using a DerSimonian and Laird random-effects model, regardless of heterogeneity levels. If the study had more than two arms in terms of antihistamine used, intervention arms were studied separately, with control group split in the analysis among the subgroups. Leave-one-out sensitivity analyses were conducted to assess the robustness of the results and to identify potential outliers. Subgroup analysis was conducted to explore the differences between different antihistamine agents.

### 2.8. Certainty of Evidence

The certainty of evidence for all key outcomes was assessed using the GRADE approach with GRADEpro GDT software. The GRADE assessment encompassed five domains: risk of bias, inconsistency, indirectness, imprecision, and publication bias.

## 3. Results

### 3.1. Search Results

The initial assessment included 310 studies retrieved from database searches (Figure 1). After removal of duplicate records, 284 studies were screened, and 10 studies were fully assessed for eligibility based on exclusion criteria. Six studies were found eligible to be included in the meta-analysis. Additionally, reference lists in those studies were searched, 119 in total. After removal of duplicates, 90 studies were screened and five were sought for retrieval. One study was excluded due to no outcomes of interest. Overall, 10 studies were included [11,12,13,14,15,16,17,18,19,20].

### 3.2. Study Characteristics

A total of 675 patients were included across the 10 randomized controlled trials (Table 1). The studies were conducted in a range of countries, including Egypt, India, Iran, Iraq, Pakistan, Vietnam, South Korea, with most originating from the Middle East and Asia. Publication years ranged from 2014 to 2025. Mean participant age across studies ranged from 18 to 22 years, and the proportion of women varied between 46.7% and 73.1%.

Overall, 345 patients received isotretinoin in combination with antihistamine, whereas 330 patients received isotretinoin alone. Three trials evaluated levocetirizine [17,18,19]; in one of these [18] levocetirizine and desloratadine were assessed in separate intervention arms. All remaining studies investigated desloratadine [11,12,13,14,15,16,20]. Treatment protocols differed substantially. Eight studies [11,12,14,15,16,17,18,19] administered isotretinoin daily, while two studies [13,20] employed an alternate-day regimen. Follow-up durations ranged from 12 to 24 weeks.

### 3.3. Meta-Analysis Results

#### 3.3.1. Global Acne Grading Scale Mean Change from Baseline

At week 4, isotretinoin + desloratadine did not yield a significantly greater mean change in GAGS than isotretinoin alone (*p* = 0.06; MD = 2.02, 95% CI –0.06 to 4.10; I^2^ = 29%) (Appendix A). Sensitivity analysis showed that removing Lee et al. [11] eliminated heterogeneity (I^2^ = 0%) without altering the conclusion (*p* = 0.17; MD = 1.27, 95% CI –0.55 to 3.09) (S2). In the levocetirizine subgroup, no between-group difference was detected, and heterogeneity was high (*p* = 0.55; MD = 1.67, 95% CI –3.76 to 7.10; I^2^ = 82%) (Appendix A). When both subgroups were pooled, the effect favored the combination (*p* = 0.03; MD = 2.10, 95% CI 0.18 to 4.03; I^2^ = 55%), although substantial heterogeneity persisted (Appendix A). In the sensitivity analysis exclusion of three studies––Lee [11], Van [12] and Pandey [18]––changed the pooled analysis to non-significant (Appendix A).

At week 8, the desloratadine subgroup again showed no significant difference (*p* = 0.25; MD = 1.71, 95% CI –1.22 to 4.65; I^2^ = 68%) (Appendix A). Removing Lee et al. [11] reduced heterogeneity (I^2^ = 12%) but left the inference unchanged (*p* = 0.51; MD = 0.62, 95% CI –1.23 to 2.47) (Appendix A). The overall pooled estimate was likewise non-significant (*p* = 0.07; MD = 2.24, 95% CI –0.19 to 4.66; I^2^ = 72%) (Appendix A). In the sensitivity analysis, exclusion of study conducted by El-Ghareeb [16] changed the results to statistically significant (*p* = 0.02; MD = 2.87, 95% CI 0.39 to 5.35; I^2^ = 68%), but with almost no impact on heterogeneity (Appendix A). Exclusion of neither study had significant effect on pooled results heterogeneity.

At week 12, the desloratadine subgroup showed a significant benefit for the combination (Figure 2; *p* = 0.0003; MD = 2.85, 95% CI 1.31 to 4.39; I^2^ = 2%). In contrast, the levocetirizine subgroup did not (Figure 2; *p* = 0.37; MD = 1.63, 95% CI –1.96 to 5.21; I^2^ = 57%). Pooled analysis favored combination therapy with no heterogeneity (Figure 2; *p* < 0.00001; MD = 2.68, 95% CI 1.60 to 3.75; I^2^ = 0%). Sensitivity tests showed no study materially affected these results. 

By week 16, the combination group achieved a significantly larger mean change from baseline than control (*p* = 0.003; MD = 2.01, 95% CI 0.68 to 3.34; I^2^ = 12%), (Appendix A). In the sensitivity analysis, removing either study conducted by Mansoor et al. [15] or Van et al. [12] changed the analysis to non-significance (Appendix A).

At week 24, a significant benefit persisted in the desloratadine subgroup (*p* = 0.02; MD = 2.47, 95% CI 0.32 to 4.63; I^2^ = 0%), and in the pooled analysis (*p* = 0.006; MD = 2.61, 95% CI 0.76 to 4.46; I^2^ = 0%) (Appendix A). Sensitivity analyses confirmed the stability of these findings.

#### 3.3.2. Side Effects

In the desloratadine subgroup, the combination therapy statistically significantly lowered the risk of acne flare-ups relative to isotretinoin alone (Figure 2; *p* = 0.0007; OR 0.36, 95% CI 0.17 to 0.76; I^2^ = 0%). There was no benefit observed in the levocetirizine subgroup (Figure 2; *p* = 0.55; OR 0.48, 95% CI 0.04 to 5.22; I^2^ = 0%). When the subgroups were pooled, the combination therapy group remained advantageous, but with significant heterogeneity (Figure 2; *p* = 0.04; OR 0.41, 95% CI 0.17 to 0.98; I^2^ = 46%). Sensitivity analyses showed that omitting the study conducted by Yosef et al. [17] reduced heterogeneity to none and changed the results to statistically significant (*p* = 0.0005, OR 0.31 95% CI 0.16 to 0.60, I^2^ = 0%) (Appendix A).

A significant benefit was observed in the desloratadine subgroup regarding cheilitis reduction (Figure 2; *p* = 0.01; OR 0.37, 95% CI 0.17 to 0.81; I^2^ = 18%). No significant difference was found in the levocetirizine subgroup (Figure 2; *p* = 0.31; OR 0.55, 95% CI 0.18 to 1.72; I^2^ = 0%). When both subgroups were combined, the analysis continued to favor the combination therapy group (Figure 2; *p* = 0.003; OR 0.41, 95% CI 0.23 to 0.73; I^2^ = 0%). However, sensitivity analysis revealed that exclusion of the study conducted by Dhaher et al. [20] rendered the result non-significant (Appendix A).

For xerosis, neither antihistamine conferred a meaningful reduction. The desloratadine subgroup showed no effect (Figure 2; *p* = 0.96; OR 0.99, 95% CI 0.55 to 1.76; I^2^ = 0%), and the levocetirizine subgroup likewise failed to reach significance (Figure 2; *p* = 0.20; OR 0.52, 95% CI 0.19 to 1.42; I^2^ = 0%). The pooled estimate was also non-significant (Figure 2; *p* = 0.50; OR 0.84, 95% CI 0.51 to 1.39; I^2^ = 0%). Sensitivity analyses confirmed that exclusion of any individual study did not impact the overall results.

Desloratadine markedly reduced pruritus (Figure 2; *p* < 0.00001; OR 0.13, 95% CI 0.06 to 0.27; I^2^ = 0%). The levocetirizine subgroup, on the other hand, showed no clear effect and exhibited substantial heterogeneity (Figure 2; *p* = 0.35; OR 0.28, 95% CI 0.02 to 4.14; I^2^ = 83%), which contributed to overall inconsistency. Nevertheless, the pooled estimate favored the combination therapy group (Figure 2; *p* < 0.0002; OR 0.27, 95% CI 0.07 to 0.43; I^2^ = 57%). Excluding the study by Yosef et al. [17] eliminated heterogeneity while maintaining the effect (*p* < 0.00001; OR 0.12, 95% CI 0.06 to 0.23; I^2^ = 0%) (Appendix A). A leave-one-out sensitivity analysis confirmed that the results were robust and stable.

#### 3.3.3. Inflammatory Lesions Count Mean Change from Baseline

At week 4, in the primary analysis, the mean change in inflammatory lesion count from baseline did not differ significantly between the isotretinoin + desloratadine group and the isotretinoin-only group (*p* = 0.23; MD 3.77, 95% CI –2.32 to 9.87; I^2^ = 62%), (Appendix A). Leave-one-out analysis identified the study by Hazarika et al. [14] as an influential outlier (Appendix A). After excluding this trial, heterogeneity dropped to 0%, and the effect in the desloratadine subgroup became statistically significant (*p* = 0.02; MD 7.79, 95% CI 1.33 to 14.26; I^2^ = 0%). In the primary pooled analysis, there was no statistically significant difference (*p* = 0.10; MD 3.63, 95% CI –0.66 to 7.92; I^2^ = 50%), (Appendix A). However, after excluding Hazarika et al. [14], the pooled estimate reached statistical significance (*p* = 0.008; MD 6.13, 95% CI 1.61 to 10.64; I^2^ = 0%), (Appendix A). Exclusion of other studies in sensitivity analysis did not significantly impact the results of the pooled analysis.

At week 8, the desloratadine subgroup showed no statistically significant effect (*p* = 0.45; MD 2.10, 95% CI –3.38 to 7.57; I^2^ = 69%), and the overall analysis was likewise non-significant (*p* = 0.16; MD 4.14, 95% CI –1.63 to 9.91; I^2^ = 76%) (Appendix A). Sensitivity analysis again identified Hazarika et al. [14] as the primary source of heterogeneity (Appendix A). Excluding this study reduced I^2^ to 0% in the desloratadine subgroup, where the effect became borderline significant (*p* = 0.05; MD 5.04, 95% CI –0.05 to 10.12), and yielded a significant overall benefit in favor of the combination therapy with low heterogeneity (*p* = 0.005; MD 6.71, 95% CI 2.03 to 11.39; I^2^ = 12%). Exclusion of other studies did not significantly alter the overall results.

At week 12, prior to sensitivity analysis, neither the desloratadine subgroup (Figure 2; *p* = 0.11; MD 5.56, 95% CI –1.20 to 12.33; I^2^ = 76%) nor the levocetirizine subgroup (Figure 2; *p* = 0.21; MD 4.87, 95% CI –2.77 to 12.51; I^2^ = 0%) demonstrated statistically significant differences. The pooled estimate showed borderline significance (Figure 2; *p* = 0.05; MD 5.19, 95% CI 0.09 to 10.29; I^2^ = 65%). Following exclusion of the Hazarika et al. [14] study, heterogeneity decreased to 12%, and a clear benefit was observed both in the desloratadine subgroup (*p* = 0.0007; MD 7.96, 95% CI 3.42 to 12.50) and in the overall analysis (*p* = 0.0006; MD 7.96, 95% CI 3.42 to 12.50) (Appendix A). Exclusion of other studies did not significantly affect the pooled results.

#### 3.3.4. Non-Inflammatory Lesion Count Mean Change from Baseline

At week 4, there was no statistically significant difference between the combination therapy groups and control groups in the desloratadine subgroup (*p* = 0.20; MD 1.84, 95% CI −0.99 to 4.66; I^2^ = 39%) (Appendix A). Similarly, no statistically significant difference was observed in the levocetirizine subgroup (*p* = 0.27; MD 17.93, 95% CI −14.06 to 49.93; I^2^ = 75%) (Appendix A). When both subgroups were combined, the overall analysis also revealed no statistically significant difference between the combination therapy groups and control groups (*p* = 0.11; MD 2.97, 95% CI −0.66 to 6.60; I^2^ = 52%), (Appendix A). In the sensitivity analysis, removal of the study conducted by Hazarika et al. [14] changed the overall effect to statistically significant without substantially reducing heterogeneity (*p* = 0.04; MD 5.22, 95% CI 0.25 to 10.20; I^2^ = 40%) (Appendix A).

At week 8, there was no statistically significant difference in non-inflammatory lesion count between the combination therapy groups and control groups in the desloratadine subgroup (*p* = 0.20; MD 0.78, 95% CI −0.42 to 1.99; I^2^ = 0%), (Appendix A). The combined analysis also showed no statistically significant difference (*p* = 0.17; MD 0.84, 95% CI −0.37 to 2.04; I^2^ = 0%), (Appendix A). Sensitivity analysis revealed that exclusion of any individual study did not significantly impact the results. 

At week 12, the combination therapy group showed a statistically significant greater reduction in non-inflammatory lesion count in the desloratadine subgroup (Figure 2; *p* = 0.003; MD 2.61, 95% CI 0.91 to 4.32; I^2^ = 11%). Sensitivity analysis, excluding the study by Hazarika et al. [14], reduced heterogeneity to 0% and maintained statistical significance (*p* = 0.00001; MD 3.10, 95% CI 1.73 to 4.46; I^2^ = 0%) (Appendix A). In contrast, the levocetirizine subgroup did not show a statistically significant difference (Figure 2; *p* = 0.32; MD 16.35, 95% CI −15.67 to 48.37; I^2^ = 76%). In the overall analysis, a statistically significant difference in favor of the combination therapy group was observed (Figure 2; *p* < 0.0001; MD 2.77, 95% CI 1.38 to 4.16; I^2^ = 2%). However, sensitivity analysis of the pooled results revealed that exclusion of the study by Dhaher et al. [20], which carried a weight of 78%, eliminated the statistical significance and introduced high heterogeneity (Appendix A).

### 3.4. Risk of Bias Assessment

Only two studies [14,18] described a computer-generated, block-random sequence with allocation concealment via sequentially numbered, opaque, sealed envelopes (SNOSE), and was therefore judged low risk in this domain (Figure 3 and Figure 4). The remaining eight trials mentioned randomization but did not provide essential details on sequence generation and/or concealment, so they were rated *some concerns* [11,12,13,15,16,17,19,20]. Nine trials were open-label for participants and treating clinicians [11,12,13,14,15,17,18,19,20]. Among them, five masked the outcome assessors, whereas the other four did not. The only double-blinded trial was conducted by El-Ghareeb et al.; therefore, in domain 2 of the RoB 2 tool, only this study was judged low risk because no deviations from the intended interventions were detected [16]. Two open-label studies [15,20] were classified high risk owing to documented, asymmetric protocol deviations or per-protocol analyses that excluded post-randomization drop-outs. The remaining seven trials raised *some concerns* [11,12,13,14,17,18,19]. Five trials retained every randomized participant and were thus judged low risk [11,12,15,17,19]. The other five had an attrition rate ≥ 5% and/or excluded participants from the analysis without adequate imputation, so they were rated *some concerns* [13,14,16,18,20]. Blinded outcome assessors and standardized measurement methods were used in five trials, leading to a low-risk judgment [13,14,16,17,18]. The remaining studies lacked full blinding for all outcomes or gave insufficient detail and were therefore rated *some concerns* [11,12,15,19,20]. None of the studies provided a publicly available protocol or statistical analysis plan, so all were rated *some concerns* for selective outcome reporting. Two studies were classified as high risk of bias [15,20], whereas the remaining eight were judged to raise *some concerns* [11,12,13,14,16,17,18,19].

### 3.5. Publication Bias

Visual inspection of the funnel plots (Appendix A) showed a symmetrical distribution of studies within the pseudo 95% confidence limits, providing no indication of publication bias. Formal testing with Egger’s regression was not undertaken in the primary analysis because fewer than 10 trials were available for each outcome.

### 3.6. Certainty of Evidence

The GRADE assessment is summarized in Figure 5. The overall certainty of evidence ranged from *moderate* to *very low* across individual outcomes. Risk of bias was the most common reason for downgrading, as all included studies were rated as either having some concerns or high risk of bias using the RoB 2.0 tool. Additionally, two outcomes were downgraded for imprecision due to wide confidence intervals or intervals crossing the line of no effect. Inconsistency was rated as serious in one outcome, primarily due to moderate heterogeneity and the influence of an outlier. Detailed reasons for each judgment are provided in the table footnotes.

## 4. Discussion

This systematic review and meta-analysis aimed to evaluate whether the addition of H_1_-antihistamines (desloratadine or levocetirizine) to isotretinoin therapy produces a clinically significant improvement in the treatment outcomes of acne vulgaris compared to isotretinoin alone. 

When treatment outcomes were measured using the GAGS, our findings suggested that combining isotretinoin with second-generation antihistamines may enhance acne treatment efficacy over time. Although there was a statistically significant difference in favor of the combination therapy at week 4, no such difference was observed at week 8. However, by week 12, a clear statistical difference emerged, with no heterogeneity (I^2^ = 0%). Further follow-up indicated that the combination therapy continued to improve GAGS scores more significantly than the control, although few studies have investigated longer-term outcomes. It is important to note that the therapeutic effects of isotretinoin typically become clinically apparent only after 8–12 weeks of treatment, with maximum response achieved after 4–6 months [21,22]. Therefore, improvements observed as early as week 4 in the combination group may primarily reflect the anti-inflammatory effects of antihistamines rather than the full pharmacologic activity of isotretinoin. The absence of a significant difference at week 8 may represent a transitional phase during which isotretinoin monotherapy begins to exert its effects, reducing the contrast between groups. By week 12, the significant difference may reflect a true synergistic or additive benefit of combination therapy. Nevertheless, mean differences across timepoints consistently ranged between 2 and 3 GAGS points, and the clinical relevance of such differences should be interpreted with caution. These findings suggest a potential time-dependent benefit of combination therapy and highlight the need for further trials to confirm these effects.

Oral isotretinoin (13-cis-retinoic acid) is an effective treatment for severe and mild-to-moderate acne lesions and helps reduce acne-related scarring. [5,23]. Although its exact mechanism of action is not fully understood, isotretinoin reduces the size and activity of sebaceous glands, indirectly lowers *Cutibacterium acnes* levels on the skin surface and within follicles, normalizes keratinocyte differentiation to prevent comedone formation, and exerts anti-inflammatory effects [24]. Despite its high efficacy, isotretinoin is associated with several common dose-dependent adverse effects that may significantly impact patient comfort and adherence [25,26]. Cheilitis occurs in approximately 90% of patients and is the most frequently reported side effect. Other common reactions include xerosis, xerostomia, nasal dryness, and photosensitivity [27]. Isotretinoin therapy may also be associated with the occurrence of flare-ups during its initial phase [28]. 

H_1_-receptor antagonists exert anti-inflammatory effects by inhibiting the release of pro-inflammatory cytokines (e.g., IL-4, IL-6, IL-13), leukotrienes, prostaglandins, and reactive oxygen species, as well as by suppressing inflammatory cell activation and mediator release [29]. In addition to their role in histamine blockade, antihistamines exert sebum-regulating effects [29]. Specifically, they may reduce squalene secretion––a biomarker of sebum––by inhibiting overexpressed histamine receptors in sebocytes, thereby lowering squalene levels [30]. 

This effect is independent of isotretinoin, which does not influence squalene production [30]. Reduced sebum release subsequently limits microcomedone formation and attenuates inflammation [7]. Those effects may be particularly useful to address side effects of isotretinoin.

Our analysis suggests that the addition of desloratadine to isotretinoin may improve treatment tolerability by reducing certain mucocutaneous side effects, particularly acne flare-ups, cheilitis, and pruritus. 

However, it should be noted that after excluding a high risk-of-bias study from the cheilitis analysis, the statistical significance was lost (*p* = 0.06); therefore, these results should be interpreted with caution. No consistent effect on xerosis was observed in the primary analyses. A meta-analysis published in 2024, which included only four RCTs, focused on pruritus and acne flare-ups in patients treated with antihistamines combined with isotretinoin compared to isotretinoin alone [10]. Despite its narrower scope, the findings were consistent with those of our analysis. 

The beneficial effect of desloratadine on inflammatory lesions appeared to be significant over time, with improvements observed consistently at weeks 4, 8, and 12. However, the inclusion of the study by Hazarika et al. [14] attenuated this effect, likely due to the very low baseline lesion count reported in that study, which contrasted markedly with the other included studies. Such conditions may have introduced a floor effect, whereby minimal disease activity at baseline limits the potential to demonstrate meaningful numerical improvement. Even if a biological effect exists, its clinical impact may not reach statistical significance when initial severity is low. Once this outlier study was excluded from the analysis, the treatment effect became more pronounced and consistent across all time points, accompanied by a substantial reduction in heterogeneity. This finding suggests that the benefits of combination therapy may be more evident in patients with higher baseline lesion counts and greater acne severity at the start of treatment.

For non-inflammatory lesions, a similar but more subtle and delayed pattern was seen. No differences were seen at earlier timepoints but by week 12 a significant benefit emerged for the combination group––again, only after the Hazarika et al. [14] study was excluded. The most likely explanation is the same: very low baseline lesion count limited the potential for improvement and masked the treatment effect in the pooled analysis. Another meta-analysis [8], which included only five RCTs, reported significant reduction both in inflammatory lesions and in non-inflammatory lesions. However, no follow-up time points were precised. 

Isotretinoin therapy may also cause hypertriglyceridemia, musculoskeletal symptoms, and laboratory abnormalities, including elevated liver enzymes and altered hematologic parameters [27]. The 2024 AAD guidelines recommend routine monitoring of liver function tests, a fasting lipid panel, and, in patients with pregnancy potential, a pregnancy test. Routine complete blood count (CBC) monitoring is no longer recommended [5]. While too few studies were available to perform a meta-analysis on these laboratory outcomes, the studies by El-Ghareeb et al. [16] and Asilian et al. [13] assessed liver enzymes, lipid profiles, and hematologic parameters, and found no statistically significant differences between the isotretinoin monotherapy and combination therapy groups. This supports the idea that adding desloratadine does not negatively impact the biochemical safety profile of isotretinoin therapy.

Combination therapies have been increasingly explored in the literature as a strategy to enhance treatment efficacy and reduce adverse effects. Currently, there are no standardized protocols for the combined use of oral isotretinoin with other therapeutic modalities, and the 2024 AAD guidelines do not provide specific recommendations regarding such combinations [5]. In specific acne subtypes such as acne fulminans, combining isotretinoin with systemic corticosteroids plays a therapeutic role in reducing severe inflammation [31]. Nonetheless, several adjunctive strategies have been explored in the literature, including photodynamic therapy [32], energy-based interventions [33,34], including laser therapy [35], and the adjunctive use of oral supplements and topical agents [36]. Of particular interest, second-generation H_1_-antihistamines such as desloratadine and levocetirizine emerge as promising adjuncts [29]. While these combinations remain outside formal guideline recommendations, accumulating evidence supports further investigation into their clinical utility for improving overall treatment tolerance.

Although this meta-analysis primarily evaluated clinical efficacy and tolerability outcomes, the observed improvements, particularly in pruritus frequency and mucocutaneous adverse events, may have important implications for patients’ quality of life. Acne vulgaris is consistently associated with psychological distress, including reduced self-esteem, social withdrawal, and increased risk of anxiety and depression [37]. The new tool of the acne-specific quality of life questionnaire, a validated patient-reported outcome measure, captures these multidimensional impacts of this particular skin disease and would be very useful if performed in these studies [38]. We believe that a reduction in visible inflammation and side effects like cheilitis or pruritus likely translates into better patient comfort, treatment adherence, and social functioning. Prior research has shown that efficacy and tolerability (e.g., side effects, cosmetic acceptability) of treatment, as well as impact on quality of life, influence treatment adherence [39]. A very important and often underestimated patient-related factor is mental health. In fact, psychiatric disorders like depression and anxiety have been demonstrated to be significant risk factors for non-adherence [40]. Therefore, the addition of antihistamines to isotretinoin, by alleviating mucocutaneous side effects, may contribute not only to enhanced clinical outcomes but also to improved psychosocial well-being. Although current guidelines do not formally recommend antihistamine co-therapy, our study suggests that incorporating antihistamine agents may be a worthwhile consideration in the management of acne. While quality of life was not systematically assessed in most of the included studies, future trials should consider incorporating validated quality of life instruments to better evaluate the holistic impact of combination therapies. Only the study by Yosef et al. [17] evaluated quality of life using the DLQI index and found higher satisfaction among patients treated with antihistamines; 64% of these patients reported being “very satisfied” compared to 36%, 24%, and 8% of control group patients who reported being “satisfied,” “slightly satisfied,” and “not satisfied,” respectively (*p* = 0.07). Although this result did not reach statistical significance, it showed a clear trend in favor of combination therapy. Given the chronic and stigmatizing nature of moderate-to-severe acne, integrating psychological screening and support into acne management may enhance long-term outcomes beyond lesion counts [3]. Therefore, quality of life should be routinely included as a relevant outcome measure in future randomized controlled trials.

## 5. Limitations

The included studies demonstrated a moderate to high risk of bias according to the RoB-2 tool. Furthermore, both intervention and control groups were generally small across all trials. The studies also differed in terms of dosing frequency and the total dosage administered for both isotretinoin and the antihistamines. Additionally, patients varied in their baseline GAGS scores and lesion counts. These factors may contribute to the observed heterogeneity in the meta-analysis, however, this reflects the current state of available evidence on combination therapy in acne treatment. 

A potential limitation of this meta-analysis is that most included studies were conducted in Asian and African populations. Some Asian subgroups show increased inflammation and skin sensitivity, which may influence the use and tolerability of adjunctive treatments like antihistamines [41,42]. Moreover, literature indicates that individuals with darker skin tones, such as Fitzpatrick types V–VI (common among African and some Asian populations), may be more prone to pigmentation changes. While these findings are frequently reported and supported by clinical observations, there is a lack of randomized controlled trials directly comparing these groups with individuals with lighter skin tones in the context of isotretinoin treatment [43,44,45,46].

These factors could help explain their more frequent use in the analyzed studies and should be considered when applying the findings to other populations.

Importantly, our findings highlight the need for future well-powered, double-blinded, randomized controlled trials with standardized dosing protocols, uniform outcome measures such as GAGS or lesion count, and stratification by baseline severity. Moreover, as most studies were conducted with per protocol analysis, therefore losing benefit from randomization, future studies should introduce intention to treat analysis. Harmonization in study design and clearer reporting––particularly regarding randomization and blinding––would substantially improve the interpretability and comparability of results across trials. Addressing these gaps will be essential for determining the true clinical value of adjunctive antihistamines in acne therapy in upcoming future research

## 6. Conclusions

This analysis shows that combining H_1_-antihistamines, especially desloratadine, with isotretinoin improves acne treatment outcomes. The combination therapy reduces common mucocutaneous adverse events such as cheilitis, pruritus, and acne flare-ups; leads to a significantly greater reduction in mean GAGS score; and further decreases both inflammatory and non-inflammatory lesion counts compared with isotretinoin monotherapy. More studies in larger populations are needed, but these results support combination therapy to optimize both clinical and patient outcomes. Future trials should also include patient reported outcome measures to better assess the psychosocial and quality of life impact of treatment.

## Figures and Tables

**Figure 1 biomedicines-13-01847-f001:**
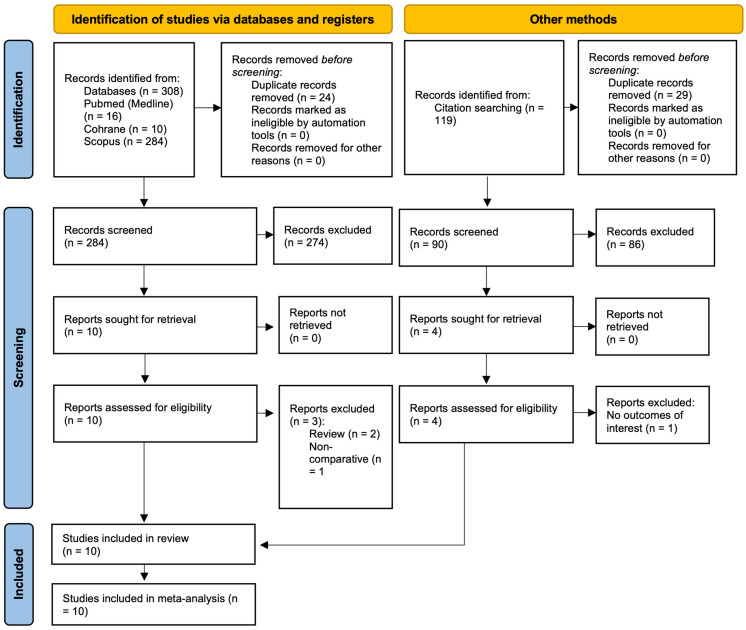
Prisma Flow Diagram.

**Figure 2 biomedicines-13-01847-f002:**
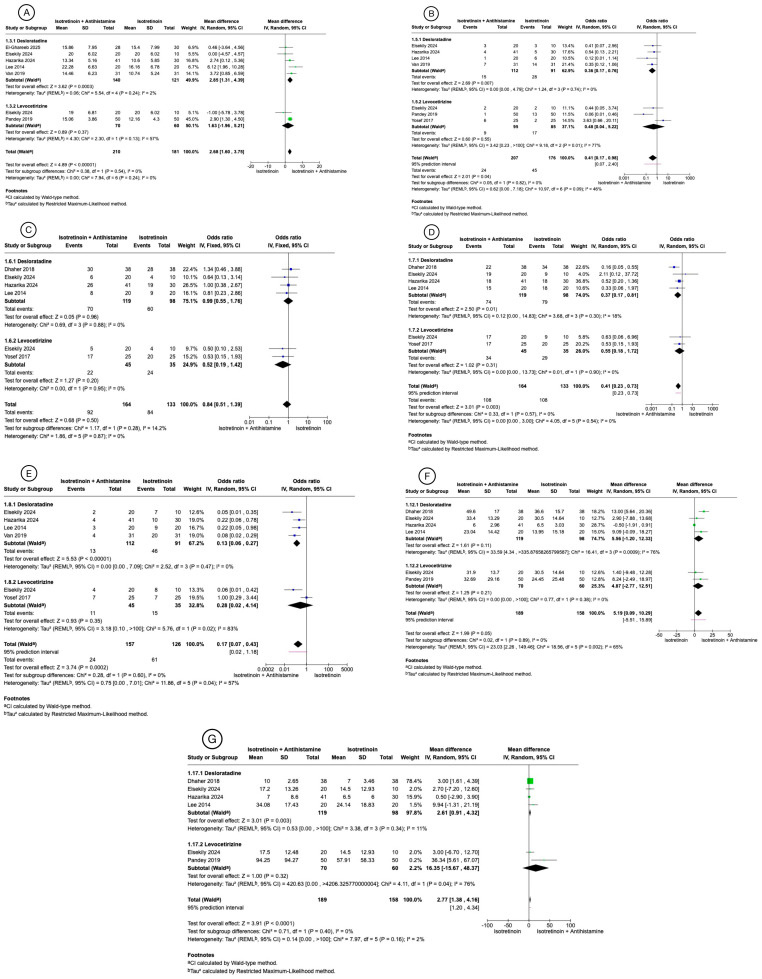
Main results of meta-analysis are shown [11,12,14,16,17,18,19,20]. (**A**) GAGS mean change from baseline to week 12. (**B**) Acne flare-ups total incidence. (**C**) Xerosis total incidence. (**D**) Cheilitis total incidence. (**E**) Pruritus total incidence. (**F**) Inflammatory lesions count mean change from baseline to week 12. (**G**) Non-inflammatory lesions count mean change from baseline to week 12.

**Figure 3 biomedicines-13-01847-f003:**
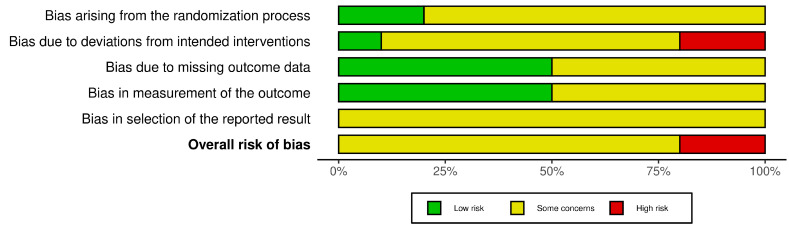
Risk of bias graph.

**Figure 4 biomedicines-13-01847-f004:**
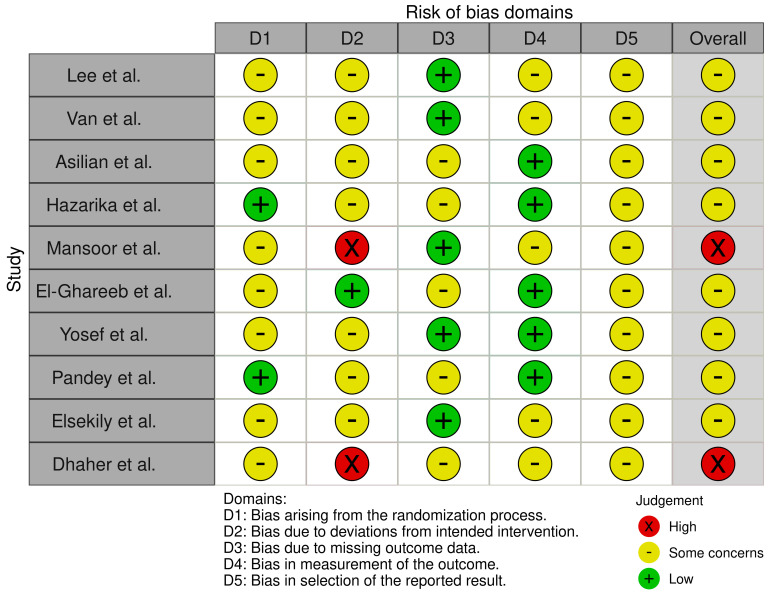
Risk of bias summary [11,12,13,14,15,16,17,18,19,20].

**Figure 5 biomedicines-13-01847-f005:**
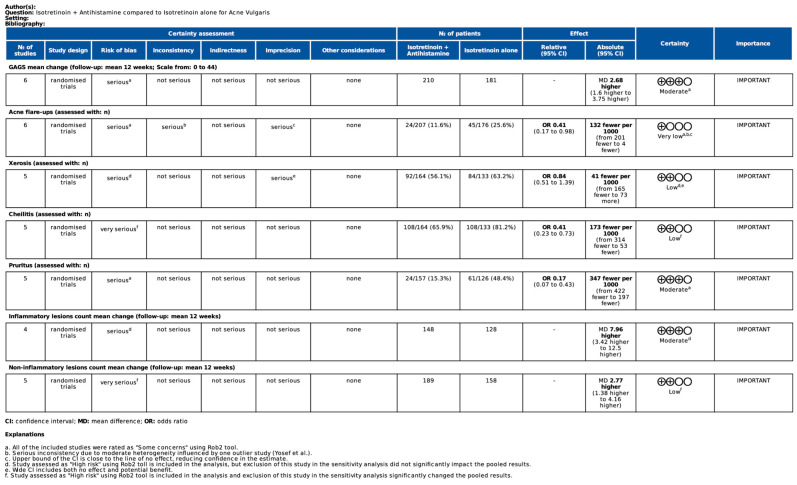
Summary of findings and certainty assessment of main clinical outcomes.

**Table 1 biomedicines-13-01847-t001:** Baseline characteristics of the included studies.

Study	Country	No. of Patients	Females (%)	Mean Age (SD)	Isotretinoin Dose	Anthistamine Dose	Follow-Ups
Lee [11] 2014	South Korea	40	60%	21.45 (2.22)	I–20 mg/day C–20 mg/day	Desloratadine 5 mg/day	Week 2, 4, 8, 12
Van [12] 2019	Vietnam	62	63.9%	21.98 (4.12)	I–20 mg/day C–20 mg/day	Desloratadine 5 mg/day	Week 2, 4, 8, 12, 16
Asilian [13] 2024	Iran	56	N/A	22.29 (2.82)	I–20 mg every other day C–20 mg every other day	Desloratadine 5 mg/day	Week 8, 16, 24
Hazarika [14] 2024	India	71	46.67%	20.25 (3.72)	I–0.3 mg/kg/day C–0.3 mg/kg/day	Desloratadine 5 mg/day	Week 4, 8, 12
Mansoor [15] 2024	Pakistan	108	73.1%	22.1 (1.26)	I–20 mg/day C–20 mg/day	Desloratadine 5 mg/day	Week 4, 8, 12, 16
El-Ghareeb [16] 2025	Egypt	58	52.5%	18.59 (0.91)	I–0.5 mg/kg/day C–0.5 mg/kg/day I–0.25 mg/kg/day C–0.25 mg/kg/day	Desloratadine 5 mg/day	Week 4, 8, 12
Yosef [17] 2017	Egypt	50	64%	19.38 (1.93)	I–0.4–0.6 mg/kg/day C–0.4–0.6 mg/kg/day	Levocetirizine 5mg/day	Week 4, 8, 12, 16
Pandey [18] 2019	India	100	66%	21.67 (4.05)	I–0.5–0.6 mg/kg/day C	Levocetirizine 5 mg/day	Week 4, 8, 12
Elsekily [19] 2024	Egypt	60	60%	21.1 (1.8)	I–0.25–1 mg/kg/day C–0.25–1 mg/kg/day	Levocetirizine/Desloratadine 5 mg/day	Week 4, 12, 24
Dhaher [20] 2018	Iraq	76	55%	19.2 (NA)	I–20 mg 3x per week C–20 mg 3x per week	Desloratadine 5 mg/day	Week 4, 8, 12

## Data Availability

Data is contained within the article or Appendix A.

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
