# Peer review of "Efficacy of Combined Oral Isotretinoin and Desloratadine or Levocetirizine vs. Isotretinoin Monotherapy in Treating Acne Vulgaris: A Systematic Review and Meta-Analysis of Randomized Controlled Trials"

_biomedicines, 2025, doi:10.3390/biomedicines13081847_

Round 1

Reviewer 1 Report

Comments and Suggestions for Authors
  • Ensure consistent terminology (for example: Use consistent terminology such as "combination therapy" or "adjunct therapy" when referring to the addition of antihistamines)
  • The conclusion can be rewritten to be clearer, more assertive to match the tone of a scientific systematic review.
  • Figures are not clear, need to be higher resolution.
  • Table 2 is not clear
  • In the abstract (conclusion), emphasize the statistically significant improvements in tolerability and GAGS score.
  • The abstract does not mention the number of studies included, sample sizes, or geographic diversity, all of which are important for assessing generalizability. Consider briefly stating these points.
Comments on the Quality of English Language
  • Improve Language and Flow

Author Response

Thank you very much for taking the time to review this manuscript. Please find the detailed responses below and the corrections highlighted in the re-submitted files.

Comments 1: “Ensure consistent terminology (for example: Use consistent terminology such as "combination therapy" or "adjunct therapy" when referring to the addition of antihistamines)”

Response 1: Thank you for pointing this out. To ensure consistency throughout the manuscript, we have standardized the terminology and now uniformly use the term “combination therapy” when referring to the use of antihistamines alongside isotretinoin.

Comments 2: “The conclusion can be rewritten to be clearer, more assertive to match the tone of a scientific systematic review.”

Response 2:  We have, accordingly, revised the Conclusion section to adopt a more concise and assertive tone that aligns with the expectations of a scientific systematic review. The updated version emphasizes key findings—including statistically significant improvements in both treatment efficacy and tolerability—and clearly states the clinical implications of combination therapy with antihistamines and isotretinoin.

Comments 3: “Figures are not clear, need to be higher resolution.”

Response 3:  Thank you for noticing. Figures are now in higher resolution, as requested. Key outcomes assessed with GRADE are now presented in one figure, the rest is now in supplementary material, as requested also by second reviewer. 

Comments 4: “Table 2 is now also in higher resolution, with short commentary below.”

Response 4:  We have made corrections regarding Table 2. It is now in higher resolution, with short commentary below. 

Comments 5: In the abstract (conclusion), emphasize the statistically significant improvements in tolerability and GAGS score.

Response 5:  We agree and have revised the abstract conclusion to explicitly highlight the statistically significant improvements observed in both GAGS scores and tolerability outcomes, particularly the reduction in acne flare-ups, cheilitis, and pruritus associated with combination therapy.

Comments 6: The abstract does not mention the number of studies included, sample sizes, or geographic diversity, all of which are important for assessing generalizability. Consider briefly stating these points.

Response 6:  Thank you for pointing it out, the abstract has been updated to include the number of randomized controlled trials (n = 10), the total number of patients (n = 675), and a brief note on their geographic distribution, which was predominantly in Asia and the Middle East. 

Reviewer 2 Report

Comments and Suggestions for Authors

1. Introduction
The Introduction section presents the purpose of the study, but there is no justification for the authors' decision to undertake this research.

2. Figures 2–5
Figures 2–5 are illegible and, in their current form, do not provide significant information.
Please move them to the Supplementary Material and replace them with a single, clearer figure that presents the key results.

3. Subsection 3.6. Certainty of Evidence
The "Certainty of Evidence" subsection contains only a table, without any description.
Please add a short comment explaining the results of the table.
Additionally, Table 2 is illegible – the layout should be improved and the title should be moved above the table (in accordance with editorial guidelines). 

4. Conclusion
The authors write:
"Darker skin types, such as Fitzpatrick V–VI, are more prone to pigmentation changes and acne-related complications..."
Please indicate how many publications and the size of the patient group this conclusion was based on.
Were the number of studies and the study population comparable across different skin types?

5. Conclusion
In the Conclusion, the authors write:
"A thorough analysis of the latest studies reveals that, despite the lack of clear recommendations, there is potential for combining antihistamines with isotretinoin."
Please specify what this potential is—for example, provide the percentage of patients who reported improvement in the analyzed studies.
The Conclusion further includes a reference to the literature:
"...future studies should also consider the newly developed quality of life questionnaire for people with acne [31]..."
Citations should not be included in the Conclusion. Please remove the reference to the literature and explain what the authors mean by "newly developed quality of life questionnaire"?

6. References
The number of cited publications is too small for the scope of the topic.
Please expand the literature review.

Author Response

Thank you very much for taking the time to review this manuscript. Please find the detailed responses below and the corrections highlighted in the re-submitted files.

Comments 1: The Introduction section presents the purpose of the study, but there is no justification for the authors' decision to undertake this research.

Response 1:  Thank you for your comment. We agree that the rationale for conducting this systematic review and meta-analysis required further clarification. Accordingly, we have revised the Introduction section to better justify the study’s purpose by emphasizing limitations of prior reviews and the need for a stratified, updated synthesis of newly available evidence.

Comments 2: Figures 2–5 are illegible and, in their current form, do not provide significant information. Please move them to the Supplementary Material and replace them with a single, clearer figure that presents the key results.

Response 2:  Thank you for noticing. We have made the corrections. The key outcomes assessed for certainty of evidence with the GRADE approach have been consolidated in Figure 2. All other outcomes have been moved to the Supplementary Material 2. The sensitivity analyses and funnel plots are now presented in Supplementary Material 3. Every subgroup and pooled analysis now includes a cross‑reference to Figure 2 or the Supplementary Material to maintain clarity.

Comments 3: Subsection 3.6. Certainty of Evidence: The "Certainty of Evidence" subsection contains only a table, without any description. Please add a short comment explaining the results of the table. Additionally, Table 2 is illegible – the layout should be improved and the title should be moved above the table (in accordance with editorial guidelines).

Response 3:  Accordingly, the requested comment has been added. We have also improved the resolution of Table 2 and moved its title above the table as instructed. Thank you.

Comments 4: The authors write: "Darker skin types, such as Fitzpatrick V–VI, are more prone to pigmentation changes and acne-related complications..."

Please indicate how many publications and the size of the patient group this conclusion was based on. Were the number of studies and the study population comparable across different skin types?

Response 4:  Thank you for your valuable and accurate thought. Although the statement is very common in the literature, it is not supported by a RCTs involving direct comparisons between individuals with lighter and darker skin types using isotretinoin. This gap in literature may have made the original claim an overstatement. 

Moreover, the evidence base for skin of color remains limited and less robust than for lighter skin types, which necessitates caution when drawing direct comparisons. This has now been revised as above.

Comments 5: In the Conclusion, the authors write:

"A thorough analysis of the latest studies reveals that, despite the lack of clear recommendations, there is potential for combining antihistamines with isotretinoin."

Please specify what this potential is—for example, provide the percentage of patients who reported improvement in the analyzed studies.

The Conclusion further includes a reference to the literature:

"...future studies should also consider the newly developed quality of life questionnaire for people with acne [31]..."

Citations should not be included in the Conclusion. Please remove the reference to the literature and explain what the authors mean by "newly developed quality of life questionnaire"?

Response 5: 

We have fully revised the Conclusion section to clarify the clinical potential of combining antihistamines with isotretinoin. We now explicitly state that combination therapy was associated with a reduction in mucocutaneous adverse events and a statistically significant improvement in GAGS scores. We have also removed the in-text citation and rephrased the sentence rekated to the acne-specific quality of life questionnaire. We wanted to highlight that the AAD has developed a new quality-of-life questionnaire specifically dedicated to patients with acne (DOI: 10.1016/j.jdin.2024.03.007). We believe this is a valuable and innovative tool that should be used in dermatology clinics as part of holistic care for patients with this condition. The revised version explains that future trials should include validated patient-reported outcome measures to better capture the psychosocial and multidimensional impact of acne, without directly citing the tool. 

Comments 6: The number of cited publications is too small for the scope of the topic.

Please expand the literature review.

Response 6:  Thank you for your comment. We have expanded the literature review and strengthened our statements with additional, relevant references.